# Brain-Type Glycogen Phosphorylase (PYGB) in the Pathologies of Diseases: A Systematic Review

**DOI:** 10.3390/cells13030289

**Published:** 2024-02-05

**Authors:** Caiting Yang, Haojun Wang, Miaomiao Shao, Fengyu Chu, Yuyu He, Xiaoli Chen, Jiahui Fan, Jingwen Chen, Qianqian Cai, Changxin Wu

**Affiliations:** 1Institutes of Biomedical Sciences, Shanxi University, Taiyuan 030006, China; yangcaiting@sxu.edu.cn (C.Y.); 13633534161@163.com (H.W.); chufengyu999@163.com (F.C.); xh19980424@163.com (Y.H.); 15035794604@163.com (X.C.); 18636833816@163.com (J.F.); 17696118526@163.com (J.C.); 2School of Medicine & Holistic Integrative Medicine, Nanjing University of Chinese Medicine, Nanjing 210023, China; shmm16@njucm.edu.cn; 3Shanghai Key Laboratory of Molecular Imaging, Shanghai University of Medicine and Health Sciences, Shanghai 201318, China

**Keywords:** brain-type glycogen phosphorylase (PYGB), glycogen phosphorylase, glycogen metabolism, pathology of diseases

## Abstract

Glycogen metabolism is a form of crucial metabolic reprogramming in cells. PYGB, the brain-type glycogen phosphorylase (GP), serves as the rate-limiting enzyme of glycogen catabolism. Evidence is mounting for the association of PYGB with diverse human diseases. This review covers the advancements in PYGB research across a range of diseases, including cancer, cardiovascular diseases, metabolic diseases, nervous system diseases, and other diseases, providing a succinct overview of how PYGB functions as a critical factor in both physiological and pathological processes. We present the latest progress in PYGB in the diagnosis and treatment of various diseases and discuss the current limitations and future prospects of this novel and promising target.

## 1. Introduction

The main composition of glucose stored in cells, known as glycogen, has a large molecular weight and is branched. Glycogen is essential for the body’s energy supply and glucose homeostasis. Glycogen metabolism, and the subsequent cellular energy balance, are important for all cells. Glycogen phosphorylase (GP) and glycogen synthase are the two enzymes involved in the allostery and covalent modification that control glycogen metabolism. As the key enzyme in the glycogen catabolism process, glycogen phosphorylase catalyzes the cleavage of glycogen’s α-1,4-glyco-sidic bonds to generate glucose-1-phosphate (Glc-1-P) units. The main source of the body’s energy is glucose-1-phosphate. Glycogen phosphorylase has been under investigation since the 1930s; it was first extracted from rabbit skeletal muscle in 1936 [1].

The three main subtypes of GP found in human tissues—brain-type glycogen phosphorylase (PYGB), liver-type glycogen phosphorylase (PYGL), and muscle-type glycogen phosphorylase (PYGM)—differ in function, structure, and tissue distribution [2]. The expression of glycogen phosphorylase isozymes has been shown to be controlled by both developmental and tissue-specific mechanisms through protein gel electrophoresis [3].

According to a chromosome-mapping analysis, the genes for muscle, liver, and brain glycogen phosphorylase were found to be located on chromosomes 11, 14, and 20, respectively. This finding suggests that different cis-acting elements were in charge of regulating the isoform-specific expression of the phosphorylase in various tissues [4]. Their expressions in human tissues vary, as seen in Figure 1. PYGM is mainly expressed in skeletal muscle, functioning to quickly mobilize and decompose muscle glycogen, providing necessary energy for muscle cell movement [5,6]. PYGL is predominantly expressed in the liver, catalyzing the decomposition of hepatic glycogen to maintain the stability of the body’s blood glucose levels [6,7,8].

In addition to playing a major role in the first step of glycogenolysis, GP also plays a specific role in specific processes. The main function of PYGM and PYGB is to participate in the production of adenosine triphosphate (ATP), which provides sufficient energy for biological processes in the cell, such as muscle contraction. PYGL produces glucose molecules to maintain the glucose levels in the bloodstream [9,10,11].

GP is a macromolecular protein polymer with multiple functional domains, and it forms a dimer when two identical subunits are in the activated state. The two states of GP are GPa (active) and GPb (inactive). GP activation involves two independent regulatory mechanisms: (1) an AMP-induced increase in GPb activity; (2) phosphorylase kinase activation by cyclic adenosine monophosphate (cAMP), which acts on GP to phosphorylate GPb to GPa [12]. In 2019, researchers elaborated on a series of reviews on the regulation mechanism of GP [3,13]. PYGB and PYGM can be regulated by both serine phosphorylation and allosteric changes, but PYGL can only be regulated by reversible phosphorylation at serine position 15 [14]. The differences in the biological effects of PYGM and PYGB are based on their biochemical properties, especially their different affinities for AMP and glucose. When the intracellular AMP levels are low, PYGB reduces its enzymatic activity and does not respond to extracellular activation signals from the phosphorylation cascade [15,16]. In addition, PYGB might be upregulated by adenosine monophosphate and downregulated by adenosine triphosphate [17,18].

The rate-determining step of glycogen degradation is catalyzed by brain glycogen phosphorylase (PYGB, also known as GPBB) encoded by the PYGB gene. The sequencing results showed that the PYGB gene is located on chromosome 20 [6]. The mRNA length of PYGB was about 4.2 kb, which was slightly larger than those of PYGL and PYGM [4]. The main reason is that the 3′ terminal untranslated region of PYGB is longer compared to those of PYGM and PYGL. According to the amino acid sequence analysis, the PYGB amino acid sequence is 83% identical to the PYGM sequence and 80% identical to the PYGL sequence. Similar to PYGL and PYGM, brain phosphorylase also exists in an active-dimer form. Remarkably, the N-terminal domains of all glycogen phosphorylases have higher levels of sequence conservation than the C-terminal domains [4]. The length of the protein predicted by the PYGB cDNA sequence is 862 amino acids, and the molecular mass is 112,917 Daltons. In contrast, PYGM and PYGL contain 841 and 846 amino acids, respectively, as seen in Figure 2. The increase in the PYGB size is entirely due to the insertion of amino acids at the C-terminal [2,4].

PYGB is primarily expressed in adult brain tissue. In addition, according to some research, PYGB is the major isomer of GP present in tumor, fetal, and heart tissues [19,20,21]. In emergencies such as ischemia, hypoxia, and hypoglycemia, PYGB catalyzes the degradation of glycogen to produce glucose-1-phosphate (G-1-P), serving as a temporary source of energy for the tissues and organisms [22,23,24,25]. Previous studies have reported that PYGB can regulate a variety of physiological and pathological processes [26,27,28,29]. It has been found to inhibit the production of reactive oxygen species (ROS) in mammalian cells and to suppress the cell apoptosis in *Escherichia coli* [29].

PYGM and PYGB colocalize in cardiomyocytes. In addition, the heart–brain ratios of PYGM and PYGB proteins and mRNAs were similar, suggesting that the coexistence of their isoforms in cardiomyocytes must be important for cardiac function [18].

An analysis of the available data shows that PYGB is involved in a number of important biological processes, especially those that require a rapid supply of energy. And we found that PYGB is highly expressed in many kinds of tissues, while the expressions of PYGM and PYGL are limited (Figure 1). There is increasing evidence suggesting that PYGB may be closely related to many human diseases, including cancer, diabetes, neurodegenerative diseases, and cardiovascular diseases. The overexpression of PYGB has been observed in various cancer types, and it has been reported that PYGB can regulate the malignant phenotypes of cancer cells [26,30,31,32,33,34]. However, the detailed molecular mechanisms by which PYGB contributes to these diseases remain elusive.

A systematic review of PYGB in diseases has not been available up until now. Here, we provide a succinct overview of how PYGB functions as a critical factor in both physiological and pathological processes. This suggests that targeting PYGB may offer new approaches and strategies for the treatment of diseases, potentially evolving into a novel therapeutic intervention strategy. In-depth research on the PYGB expression mechanism will provide new ideas and directions for the diagnosis, treatment, and prognosis of diseases.

## 2. PYGB in Cancer

Cancer is a leading cause of death worldwide, and one of its features is the reprogramming of metabolism [35]. Numerous studies have shown that glycogen metabolism plays a key role in the progression of cancer, and inhibiting glycogen metabolism may be a useful anti-cancer treatment strategy [36,37]. There is evidence that PYGB is associated with the progression of varied human malignant tumors, regulating the proliferation, invasion, and apoptosis of cancer cells, and other biological characteristics. Research by Favaro et al. has demonstrated that glucose maintains proliferation by utilizing GP and prevents premature aging in cancer cells [37]. PYGB was first found to be highly expressed in gastric cancer cells and localized in the nucleus [38]. Emerging evidence hints that PYGB is involved in the tumorigeneses of different types of cancer, such as osteosarcoma [17], breast cancer [26], prostate cancer [29], colorectal cancer [33], gastric cancer [39], ovarian cancer [40], and non-small-cell lung cancer [41], as well as hepatocellular cancer [42]. Studies have also indicated that the imbalance in PYGB may be used as a metabolic target for clinical applications.

Due to the high incidence, rapid development, and poor diagnosis of cancer, the discovery of diagnostic markers for cancer and more effective treatments is of paramount importance. Here, we explore the role of PYGB in cancer and propose that PYGB is a promising target for the development of new medications. Because the PYGB levels are highly expressed in a variety of cancers, as we show in Figure 3, and elevated PYGB expression levels are associated with poorer survival in patients, this process also appears to play a central role in cancer progression.

### 2.1. PYGB in Ovarian Cancer

Ovarian cancer is one of the most common gynecologic cancers, with high morbidity and mortality in women. Studies have shown that the high expression of PYGB in ovarian cancer is positively correlated with a poor prognosis for ovarian cancer patients. Xenograft tumor formation further demonstrated that the knockdown of the PYGB gene significantly inhibited the proliferation, invasion, and migration of ovarian cancer cells, suppressing ovarian cancer tumorigenesis. Knocking down PYGB also significantly inhibited the expressions of Wnt3a, β-catenin, APC, and Cyclin D1 in ovarian cancer cells, suggesting that, through the Wnt/β catenin signaling pathway, PYGB might regulate the progression of ovarian cancer. In terms of the mechanism, the results showed that miR-133a-3p targeted PYGB and combined with the 3′-UTR of PYGB to inhibit the development of ovarian cancer [40].

These studies have demonstrated that miR-133a-3p may play a critical role in the occurrence and development of ovarian cancer through the Wnt/β-catenin signaling pathway, and that PYGB could be a key diagnostic marker and therapeutic target for the treatment of ovarian cancer.

### 2.2. PYGB in Gastric Cancer

Gastric cancer (GC) is considered one of the most common gastrointestinal malignancies with a high mortality rate.

The study by Shimada et al. has shown that PYGB is highly expressed in well-differentiated gastric adenocarcinoma and in the proliferative zone of intestinal metaplasia, suggesting that PYGB may be involved in the formation of early gastric cancer [43].

The expression level of PYGB in intestinal-type gastric cancer is significantly higher than that in the diffuse type. The positive proportion of IM in the intestinal type is obviously higher than that in the diffuse type. The relationship between intestinal metaplasia (IM) and PYGB in proliferating cells and intestinal-type carcinoma is closer than that of the conventional subtype [44].

The detection of the PYGB expression levels in intestinal metaplasia cells in early-gastric-cancer tissues and further statistical analysis show that the positive rate of PYGB expression in intestinal metaplasia cells in early-gastric-cancer tissues is significantly higher in the multi-primary-lesion population than in the single-lesion population, suggesting that the detection of PYGB expression in intestinal metaplasia cells in locally resected early-gastric-cancer lesion tissues has a certain predictive value for judging whether multiple lesions still exist [45].

In the gastric cancer cell line SNU16, the protein KIAA1199 upregulates PYGB activity, enhances cell glycogenolysis, and promotes cell proliferation [46].

Studies have found that the expression of PYGB in human gastric cancer tissues is significantly elevated and positively associated with the clinical–pathological characteristics of gastric cancer patients. The results point out that, via the Wnt/β-catenin signaling pathway, PYGB regulates the proliferation, migration, invasion, and EMT of GC cells. The depletion of PYGB also inhibits tumor growth in xenograft tumors and lung metastasis [39].

These findings shed light on PYGB as a novel promising therapeutic target that can be used to unravel the pathogenesis of GC for the design and development of drugs targeting to gastric cancer, and to provide new insights into the development of new strategies for clinical diagnosis and treatment.

### 2.3. PYGB in Hepatocellular Carcinoma

Liver cancer is the most common fatal malignant tumor worldwide. More than 90% of all cases of liver cancer are hepatocellular carcinoma (HCC).

In HCC tissues, PYGB was specifically stained, indicating that PYGB may be a potential biomarker for the diagnosis of HCC [42]. Studies have shown that PYGB is highly expressed in HCC tissues, and that its overexpression is related to the invasive tumor phenotype and poor prognoses of HCC patients [47].

These results suggest that PYGB may participate in the progression of HCC by enhancing the invasiveness of HCC cells and the EMT process. In terms of the mechanism, miR-101-3p post-transcriptionally inhibits the expression of PYGB by binding to the 3′-UTR of PYGB, regulating the tumorigenesis and metastasis of HCC. The overexpression of PYGB promotes the tumorigenesis and metastasis of HCC cells, indicating that PYGB plays a key role in the growth and metastasis of HCC and may be a new biomarker to improve the prognoses of patients with HCC [48].

### 2.4. PYGB in Pancreatic Cancer

Philips et al.’s research shows that sensitivity to GS correlates with reduced levels of PYGB. They cultured the 2-DG (2-deoxy-D-glucose)-resistant pancreatic cancer cell line in a glucose-deprived environment for 72 h, and the results showed that the protein and mRNA levels of PYGB were reduced and the cell mortality was increased significantly compared with the primary cells. This result suggests that the 2-DG-resistant cell line is more sensitive to GS than the primary cell line, which coincided with the decrease in both the PYGB levels and glycogenolysis in the 2-DG-resistant cell line [28]. It is also suggested that the expression of PYGB can promote the decomposition and utilization of glycogen by pancreatic cancer cells, improving the viability of pancreatic cancer cells in a glucose-deficient environment. This indicates that the sensitivity to GS can be increased by inhibiting PYGB in the parental cell line [28].

This year, a new study evidenced that the m6A methylation of PYGB promoted the tumorigenesis of pancreatic ductal adenocarcinoma (PAAD) through NF-κB signaling, mediated by METTL3 [49]. These findings provide a potential therapeutic target for the treatment of pancreatic cancer.

### 2.5. PYGB in Colorectal Cancer

PYGB has also been investigated as a diagnosis biomarker in human colorectal cancer. Tashima et al., through immunohistochemical research, have shown that PYGB is commonly expressed in colorectal cancer tissues [33]. PYGB is expressed in colonic adenomas with a high grade of dysplasia, and the tumor and is frequently expressed in the transitional mucosa of colorectal cancer without an adenoma component. A statistical analysis showed an excellent positive correlation between the expression of PYGB during the adenoma–carcinoma sequence (ACS) and increased dysplasia, whereas no PYGB expression was seen in the normal human large intestine away from the cancer foci. This indicated that PYGB could be used as a biomarker for the diagnosis of colorectal cancer.

In addition to the “adenoma-carcinoma sequence” (ACS), “de novo” carcinogenesis is another pathway leading to sporadic colorectal cancer. This study revealed that the multifocal expression of PYGB mainly appeared in the colorectal mucosa adjacent to the “de novo” carcinoma, and that the PYGB foci may serve as potential preneoplastic lesions of “de novo” colorectal carcinogenesis. Studies have demonstrated the potential for premalignant foci with frequent p53 gene mutations in colorectal transitional mucosa. This suggests that PYGB-positive foci might be a candidate for early preneoplastic lesions, and it is expected to contribute to in-depth studies on the pathogenesis of “de novo” colorectal cancer [31].

### 2.6. PYGB in Prostate Cancer

Prostate cancer, a common malignancy occurring in the prostate epithelium, is the second leading cause of cancer-related mortalities in older men globally.

According to the clinicopathological data, the researchers showed that the expression level of PYGB in prostate cancer tissues was significantly enhanced. In addition, they also demonstrated that the expression of PYGB was related to the malignancy degree of the prostate cancer. The analysis of the clinicopathological data also confirmed that PYGB was related to the differentiation degree and TNM stage of the prostate cancer tissues. In short, PYGB is upregulated in prostate cancer tissue and related to disease progression, suggesting that it may become an important target for the pathological diagnosis and treatment of prostate cancer and a prognostic indicator.

In addition, studies have confirmed that PYGB silencing promotes apoptosis, inhibits cell viability, and increases the cell ROS content. PYGB silencing promoted apoptosis by increasing the expressions of cleaved-PARP, caspase-3, and Bax and decreasing the expression of Bcl-2. The study also revealed that PYGB silencing significantly increased the expression of NF-κB and decreased the expression of Nrf2, thereby affecting the NF-κB/Nrf2 signaling pathway in the cells [29].

These findings may provide a new research focus for understanding the pathogenesis of prostate cancer, and contribute to the development of the diagnosis and treatment of prostate cancer.

### 2.7. PYGB in Osteosarcoma

Osteosarcoma is the most common malignant bone tumor, and it mainly occurs in children and teenagers with a poor prognosis and often leads to metastatic disease.

Studies have shown that PYGB has a higher expression level in osteosarcoma tissues, especially in the human osteosarcoma cell lines MG63 and HOS, compared to bone cysts. Knocking down PYGB will inhibit the malignant phenotypes of cancer cells. In MG63 and HOS cells transfected by PYGB siRNA, apoptosis was induced by the Bcl/Caspase and cyclin-dependent kinase (CDK)-1 signaling pathway [17].

Therefore, it can be said that PYGB is a potential marker for the early clinical diagnosis and treatment of osteosarcoma, providing a new method for its treatment.

### 2.8. PYGB in Non-Small-Cell Lung Cancer (NSCLC)

Globally, lung cancer is one of the most common malignant tumors, and non-small-cell lung cancer (NSCLC) is the most common type of lung cancer, accounting for approximately 85% of all lung cancer cases.

Nuclear glycogen metabolism in NSCLC was originally discovered in the early 1940s, but its role in cellular physiology remained elusive [50].

Schnier et al. found that the non-small-cell lung cancer cell line A549 only expressed PYGB, while PYGM and PYGL basically did not express. When the allosteric inhibitor CP-91149 of GP was added to A549 cells, the glycogen level was significantly increased, cell proliferation was inhibited, and cell apoptosis was increased [51]. These results indicated that PYGB plays an important role in glycogen utilization and the proliferation of lung cancer cells.

Via immunohistochemistry, Ki et al. detected the high expression of PYGB in NSCLC tissues. The survival analysis showed that the PYGB expression was associated with the prognoses of patients with non-small-cell lung cancer, and that patients with high PYGB expressions had poor prognoses [41].

Gentry et al. indicated that malin (an E3 ubiquitin ligase) was not targeted to degrade PYGB. But it was later found that malin can ubiquitize PYGB, indicating that PYGB is a substrate for malin. When malin and PYGB are co-expressed in cells, PYGB will be translocated from the cytoplasm to the nucleus rather than instead of being degraded by proteasome. On the contrary, co-expression with point-mutation malin abolishes the localization of PYGB in the nucleus. In addition, it was found that the overexpression of malin increased the endogenous GP in the nucleus, suggesting that PYGB translocation is part of normal cellular physiology and is controlled through malin-mediated ubiquitination. Detected via the TCGA database and an IHC analysis of patient specimens, compared to normal tissues, the mRNA and protein levels of PYGB in cancer tissues were significantly increased. In addition, PYGB is mainly located in the cytoplasm in cancer tissues, while 12% in normal tissues is located in the nucleus, on average [52]. This study uncovers a previously unknown role for glycogen metabolism in the nucleus and elucidates another mechanism by which cellular metabolites control epigenetic regulation.

In two recent studies, the researchers demonstrated that the PYGB expression was upregulated in NSCLC tissues, predicting a poor prognosis in NSCLC patients [53,54]. The expression of PYGB was positively correlated with the TNM stage, positive lymph node metastasis, and a poor prognosis in NSCLC patients. The overexpression of PYGB promotes cell proliferation, migration, and invasion and inhibits apoptosis. Moreover, the knockdown of PYGB significantly reduces the viability, proliferation, migration, and invasion of NSCLC cells, inducing their apoptosis, which may be related to the link between PYGB and the Wnt-β-catenin signaling pathway [53].

Mechanistically, the overexpression of PYGB could activate the phosphoinositide 3-kinase (PI3K)/protein kinase B (Akt) signaling pathway, while the knockdown of PYGB inhibits the PI3K/AKT signaling pathway [54].

These findings have unveiled the role of PYGB in the progression of NSCLC and its potential as a new biomarker and potential molecular therapeutic target, potentially providing a new basis for the diagnosis and treatment of NSCLC.

Based on the TCGA-LUAD cohort, 25 genes affecting OS were screened via univariate Cox regression analysis. In order to obtain a high-accuracy prognosis model, 16 NRGs were obtained via LASSO regression analysis, and, finally, 6 independent prognosis genes (PYGB, IL1A, IFNAR2, BIRC3, H2AFY2, and H2AFX) of LUAD were obtained via multivariate Cox regression analysis. Among them, the risk coefficient of PYGB is 0.4677, and the hazard ratio is 1.596 (*p* < 0.001); therefore, PYGB is a risk gene (HR > 1). At the same time, the study once again confirmed the high expression of PYGB in LUAD through the TARGET GTEx dataset of TCGA and a clinical cohort. This suggests that we can predict the prognosis and tumor immunity of LUAD through the potential signature consisting of six NRGs, including PYGB [55].

Moreover, the smoking-related pathological subtype of non-small-cell lung cancer (NSCLC) is named lung squamous-cell carcinoma (LUSC), which accounts for a high proportion of all cases of lung cancer. The researchers identified eight core genes with prognostic implications and constructed an eight-gene signature in the LUSC dataset. These helped to distinguish the survival risk of patients with LUSC and have robust prognostic value. These genes include PYGB, and the low-level amplification of PYGB may be related to the epigenetic mechanism of smoking for tumor occurrence. Thus, PYGB is more likely to be involved in the occurrence and development of LUSC related to smoking status and may serve as a potential biomarker for the diagnosis and prognosis of smoking-related LUSC [56].

Another group of researchers integrated glycolysis-related genes (GRGs) and immune-related genes (IRGs) and constructed a systematic glycolysis–immune score (GIS) model, which contains two GRGs (PYGB and MDH1) and three IRGs (TSLP, SERPIND1, and GDF2). Among them, the gene coefficient of PYGB is 0.20604779. The GIS model showed a stable prognostic efficacy in different datasets and clinical subgroups of LUSC, and it can be used to quantitatively estimate the prognosis of patients with LUSC and guide chemotherapy and immunotherapy decisions [57].

### 2.9. PYGB in Breast Cancer

As early as 2002, the differential expression of PYGB was confirmed, and PYGB was found to be downregulated in HER2/neu-positive breast cancer cells both in vitro and in vivo [58].

In their 2019 study, Altemus et al. found that breast cancers utilize hypoxic glycogen stores via PYGB to stimulate metastatic phenotypes. They reported that breast cancer cells significantly increase the glycogen stores in response to hypoxia. Although breast cells have increased glycogen storage under hypoxic conditions, PYGB-promoted glycogen utilization affects the migration and invasion of cancer cells. They found that knocking down PYGB inhibited the utilization of glycogen in breast cancer cells. The results showed that the PYGB knockdown and its resulting inhibition of glycogen utilization resulted in the significantly reduced wound-healing and invasion potential of the cells. These findings indicate that PYGB is a potential new target for reducing the metastasis and invasion of breast cancer, which has potential application value in the prevention of metastasis in breast cancers [26].

### 2.10. PYGB in Renal-Cell Carcinoma (RCC)

Takashi et al. determined that the tissue concentration of PYGB in renal-cell carcinoma was significantly higher than those in the renal cortex and medulla. The phosphorylase level of clear-cell tumors was slightly higher than that of granular-cell tumors. These findings indicated an enhanced tissue level of PYGB in renal-cell carcinoma cells [59].

However, a recent study has demonstrated that glycogen metabolism is not necessary for the proliferation of RCC in vitro or for the growth of xenografts in vivo. Glycogen breakdown was prevented by inhibiting PYGB and PYGL without affecting the cell viability. Other experiments also showed that there were no significant alterations in the tumor size or content when the glycogen metabolism was changed [60]. Therefore, the biological significance of the increased PYGB concentration in RCC tissues and the pathological background of glycogen deposition in clear-cell RCC need further investigation.

### 2.11. PYGB in Inverted Urothelial Papilloma (IUP)

Inverted urothelial papilloma (IUP) is an uncommon tumor, usually benign, but it is a precursor disease of urothelial cancer. Histological overlapping between papillary urothelial carcinoma (PUC) and IUP may frequently lead to misdiagnosis.

A recent study analyzed IUP, PUC, and normal urothelium (NU) based on proteomics and further verified via immunohistochemistry that PYGB is a promising biomarker to distinguish IUP and PUC, with a good application prospect in the routine diagnosis of IUP [61].

### 2.12. PYGB in Oral Squamous-Cell Carcinoma (OSCC)

Oral squamous-cell carcinoma (OSCC) has high mortality rates and is closely related to lymph node metastasis. Through a multi-omics analysis of data in public databases, 11 metastasis-related “hub proteins” transported by EVs, including PYGB, were identified, and their low abundance was found to be relative to the survival of cancer patients and reduced tumor invasiveness, suggesting that PYGB might be a candidate molecule for prognostic marker of OSCC [62].

### 2.13. PYGB in Glioblastoma Multiforme (GBM)

Glioblastoma multiforme (GBM) arises from the malignant transformation of astrocytoma and is the most common and aggressive malignancy of the central nervous system (CNS). By using a combination approach of biochemistry and protein studies, researchers investigated the interaction and action mechanism of 2,3-benzodiazepin-4-one, which was the blood–brain barrier permeability, towards PYGB in U87MG cell lines of glioblastoma multiforme, as well as the ability of 2,3-benzodiazepin-4-one to interfere with the activity and expression of PYGB, and the ability to inhibit cell growth and the cell cycle.

The results highlighted PYGB as a potential therapeutic target for U87MG of GBM and indicated that 2,3-benzodiazepin-4-one can be used to develop anti-cancer drugs for glioblastoma because it can negatively regulate glucose uptake and metabolism (the “Warburg effect”) [63].

## 3. PYGB in Cardiovascular Disease

### 3.1. PYGB in Ischemic Myocardial Injury 

In contemporary society, cardiovascular disease seriously endangers people’s health. Coronary atherosclerotic heart disease (CHD), sometimes called ischemic heart disease, refers to the coronary atherosclerosis caused by myocardial ischemia, hypoxia, and heart disease. Acute coronary syndrome (ACS) is a serious type of coronary heart disease and a common serious cardiovascular disease in the clinic, including acute myocardial infarction (AMI), unstable angina pectoris (UA), etc. Most of the mortality related to AMI occurs within the first hour of symptom onset [64]. Therefore, the early detection of MI is very important to reduce the morbidity and mortality related to coronary artery disease.

The prerequisite for the successful treatment of patients and for minimizing myocardial cell injury is early and accurate diagnosis, especially with the development of thrombolytic therapy. Biochemical markers for the auxiliary diagnosis of myocardial injury are crucial indicators. Biochemical markers of myocardial injury refer to changes in the membrane integrity and permeability after myocardial cell injury, leading to the escape of macromolecular substances into the blood circulation. These macromolecular substances are known as myocardial-injury biochemical markers, including creatine kinase (CK-Mb), glycogen phosphorylase isoenzyme (PYGB), myoglobin (MB), cardiac troponin (cTn), etc. PYGB is sensitive to myocardial ischemia and hypoxia, and it may become an indicator for the early diagnosis of ACS, filling the deficiencies of the traditional markers of myocardial injury.

PYGB is not only present in brain tissue but also in myocardial cells. It is a key enzyme for glycogen decomposition in the oxidative phosphorylation of myocardial cells. Under physiological conditions, PYGB and glycogen in myocardial cells bind to the sarcoplasmic reticulum, forming a macromolecular sarcoplasmic reticulum–glycogen degradation complex [65]. The degree of PYGB binding to the complex mainly depends on the metabolic state of the myocardial cells, depending on the oxygen and blood supply state of the myocardium. It is highly sensitive and specific to the acceleration of glycogen cleavage in the sarcoplasmic reticulum structure of myocardial cells caused by ischemia and hypoxia.

The release of PYGB mainly depends on the degradation of glycogen, and the degradation process is jointly catalyzed by the phosphorylated active form (GPa), the non-phosphorylated active form (GPb), and the AMP-dependent form of glycogen phosphorylase. PYGB combines with adenosine phosphate (AMP) and undergoes an allosteric effect after phosphorylation, transforming GPb with lower activity into GPa with higher activity, thereby regulating its activity.

When cardiomyocytes suffer from hypoxia, due to the rapid depletion of available energy substances, such as ATP and phosphocreatine, ADP and inorganic phosphate increase, promoting the activity of phosphorylase. Phosphatase kinase can convert GPb into GPa, and glycogen is degraded to generate G-1-P. PYGB changes from the complex particle form to the soluble form, freely flowing inside and outside the sarcoplasmic reticulum, thereby forming a high GP concentration gradient inside and outside the sarcoplasmic reticulum. Hypoxia is usually accompanied by the increased permeability of the cell membrane, allowing PYGB to enter the extracellular fluid from around the sarcoplasmic reticulum, causing an increased concentration of PYGB in the blood. PYGB increases within 1–4 h of chest pain onset and can be cleared from the blood circulation within about 24–72 h. Only in severe myocardial ischemia that the accelerated glycogen decomposition and increased membrane permeability of cardiomyocytes coexist, will GP be released [66].

#### 3.1.1. Acute Myocardial Infarction (AMI)

The clinical data showed that CK-Mb, MB, cTnT, cTnI, and PYGB each have advantages and disadvantages in the diagnosis and treatment of AMI. CK-Mb is considered the best standard for the diagnosis of AMI, with high sensitivity. MB is also a sensitive indicator of myocardial damage, appearing earlier than CK-MB but with slightly poorer specificity. CTnT and cTnI are relatively specific indicators of myocardial damage. CTnI is more specific for the myocardium and is the indicator with the longest blood retention time found at present, playing an important role in the diagnosis of AMI and the condition monitoring and prognosis monitoring of patients with unstable angina pectoris [67].

According to the report by Rabitzsch et al., the myocardial protein markers MB, CK-Mb, cTnT, and cTnI were all increased in 17 patients with AMI who received thrombolytic treatment, but the increase in PYGB was the most significant [68]. They indicated that PYGB is the most sensitive marker within 4 h of the onset of chest pain; therefore, the rapid measurement of PYGB may be crucial in the early diagnosis of AMI [69,70].

PYGB is thus a good early marker reflecting myocardial ischemia in the pathological process of coronary artery disease.

#### 3.1.2. Unstable Angina Pectoris (UA)

The clinical application of PYGB is not only limited to traditional myocardial infarction.

In all patients, acute myocardial infarction was ruled out. According to the research by Mair and Rabitzsch et al., in patients with unstable angina and transient ST-T alterations, the PYGB concentration was the only marker that was significantly increased above its discriminator value. These results suggest that PYGB can reflect the degree of myocardial damage, which is particularly important for early risk stratification in patients with UA. The early release of PYGB into the blood may help identify high-risk patients with UA upon admission to an emergency department and guide decisions about patient management [67,69,71].

The results of the sensitivity, specificity, and receiver-operating characteristic (ROC) curve demonstrated that PYGB outperformed other MI biochemical indicators in the early diagnosis of acute coronary syndrome (ACS) [67,69]. Therefore, close observation and dynamic monitoring should be conducted for changes in the ECG and slightly higher PYGB to avoid misdiagnosis or missed diagnosis.

Huang et al. monitored the time-phase changes of PYGB in patients with UA and synchronously observed CK-MB, CK, etc. There was no significant difference between the two groups. PYGB-positive patients accounted for 75% of patients with UA, and there was a correlation with their clinical conditions [72]. Therefore, the PYGB assay provides a simple, reliable, and economical method for clinically identifying high-risk populations in patients with UA.

Peetz et al. showed that, when measured within the initial 6 h, PYGB was more sensitive and specific than myoglobin and the CK-MB mass. Elevated PYGB levels were also found (at 3 h) in 93.9% of patients who received the final diagnosis of unstable angina [73]. Thus, PYGB may be a marker of ischemia or infarction.

In 2012, the results showed that PYGB could improve the early detection rate of AMI with high diagnostic accuracy. It is particularly useful for distinguishing between AMI and UA within 3 h after admission in patients with chest pain for <3 h [74].

#### 3.1.3. Coronary Artery Bypass Grafting (CABG)

Coronary artery bypass grafting increases the concentration of PYGB in the blood. Mair et al. showed that, in 16 patients with uneventful coronary artery bypass grafting, PYGB reached its peak at 4 h and decreased to its baseline values within 20 h. Compared with CK-MB activity, the peak value of the PYGB concentration is more correlated. For emergency CABG patients, because the half-life of PYGB is shorter than that of CK-MB activity, PYGB is more suitable than CK-MB in the perioperative laboratory monitoring of myocardial ischemia in patients undergoing emergency CABG. As the direct evidence of myocardial ischemic injury, its release mainly depends on myocardial ischemia and hypoxia, determining the PYGB mass concentration as a sensitive marker of perioperative myocardial injury in patients undergoing coronary artery bypass grafting [67,75].

However, surprisingly, Mion et al. showed in their research population that the diagnostic efficacy of troponin alone was greater than that of the combination of troponin and PYGB, and that the combination of PYGB and cTnI did not improve the detection rate of ACS [76].

In 2011, a study published by Meune et al. also showed that PYGB has limited incremental value in the detection of patients with non-ST-elevation ACS and failed to reflect the incremental diagnostic value of PYGB relative to troponin in ACS or MI [77].

In 2012, Lippi et al. published a meta-analysis of PYGB in the diagnosis of acute MI [78]. PYGB cannot be recommended as an independent indicator for the diagnosis of AMI because it does not meet the current requirements for an efficient diagnosis of AMI, and the combination of PYGB and troponin for the diagnosis of AMI requires further study in the future.

Besides its diagnostic value, PYGB has also been evaluated as a prognostic marker of AMI patients. A report by Lillpopp et al. pointed out that the PYGB measured at the time of admission of suspected ACS patients can provide information for the medium-term prognosis [15]. However, this was different from the previous results of McCann et al. [79], who demonstrated that PYGB did not provide independent prognostic information in patients with acute ischemic chest pain.

According to the research of Dobric and his colleagues [80], the PYGB plasma concentration increased in 46 patients with known coronary anatomies undergoing an exercise stress echocardiography test (ESET), and it was not related to inducible myocardial ischemia but to the exercise load/duration. After that, their team published a detailed summary about PYGB in myocardial infarction, concluding that [81] PYGB cannot be recommended as a diagnostic marker of AMI, neither as an independent indicator nor as an addition to troponin, which needs further investigation.

The study by Yarana and colleagues demonstrated that in a doxorubicin-induced cardiac injury mouse model, extracellular vesicles released by cardiomyocytes contain the protein biomarker of early cardiac injury, PYGB. Their analysis of the proteomic data showed that doxorubicin-treated EVs clearly contained PYGB. PYGB can be used as a new fluoroscopic marker for doxorubicin-induced heart injury to assess the risk of heart injury in patients after chemotherapy [82,83].

Recently, a meta-analysis of PYGB showed that it had moderate diagnostic accuracy for myocardial infarction (MI) and was superior to CK-MB and myoglobin. However, whether PYGB can replace hsCTn, especially in the early stages of symptom development, is still unclear, and more solid evidence is needed to prove that PYGB is a standard cardiac biomarker [84].

### 3.2. PYGB in Acute Ischemic Stroke (AIS)

The concentration of PYGB in the brain is approximately 50% higher than that in the heart [85]. Brain glycogen accounts for 0.1% of the total brain weight and is consumed very rapidly during cerebral ischemia to meet the increasing demand for ATP [86].

It has been demonstrated that PYGB can be used as a potential candidate marker for AIS [87]. First, PYGB increased very early (as early as 90 min) after AIS. The PYGB levels were above the upper reference limit within the first 4.5 h of stroke onset. Therefore, PYGB can be used as a screening indicator for the early diagnosis of suspected stroke. And the PYGB levels continue to rise for at least 48 h, allowing clinicians to establish a diagnosis of cerebral ischemia in patients with recent neurological symptoms.

PYGB has been demonstrated to be a sensitive indicator for the diagnosis of AIS; however, whether it is a specific indicator in the brain needs further study. High concentrations of PYGB are also present in the human heart [85], and concurrent myocardial injury may result in the leakage of PYGB from the myocardium, causing a false-positive elevation for PYGB [75].

A recent study showed that the accumulation of glycogen is closely related to the development of I/R injury subjected to transient cerebral ischemia. PYGB plays an important role in glycogen accumulation-related neuropathy. The PKA-PhK-GP cascade is involved in the reprogramming of glycogenolysis during recanalization after ischemic stroke and reoxygenation.

In addition, compared with those of PYGM and PYGL, the expression of PYGB is preferentially affected, but it is not certain whether only the activity of PYGB decreases during reperfusion. There is no evidence that the PKA-PhK pathway only regulates PYGB and not PYGM or PYGL. Therefore, we speculate that PYGB’s vulnerability to I/R is mainly due to its large proportion (91.7%) in GP mRNA isoforms in cultured astrocytes.

Insulin can also activate the PKA-PhK-GP pathway, and the neuroprotective effect of insulin can be weakened by PYGB knockdown, which may be the reason for the insulin-mediated recovery in the acute and subacute phases after stroke. Therefore, when the level of glycogen has increased significantly during reperfusion, insulin in the brain is more likely to act as a coordinator to maintain glycogen metabolic homeostasis rather than simply as an accelerator of glycogen synthesis [88].

### 3.3. PYGB in Brain Ischemia

In recent years, Zhang’s group revealed a novel PYGB inhibitor involved in glucose metabolism, which has better inhibitory activity against PYGB and a potential therapeutic effect on brain ischemia. They confirmed the inhibitor’s protective effect on astrocyte H/R injury by targeting PYGB, suggesting that PYGB could be an effective therapeutic target for ischemic hypoxic diseases [89,90].

### 3.4. PYGB in Preeclampsia (PE)

During pregnancy, the cardiac function and maternal oxygen levels change significantly and play a key role in adverse pregnancies, including preeclampsia (PE) and small for gestational age (SGA) [91]. Preeclampsia and coronary syndrome have common etiological and pathophysiological features. The pathophysiological changes in preeclampsia, namely, ischemia and endothelial dysfunction [92], may lead to irreversible cardiovascular damage. As a result, women with these adverse pregnancies, including hypertension of pregnancy, are at an increased risk of long-term cardiovascular disease [93]. PYGB has been proposed as a potential biomarker to detect these adverse pregnancy outcomes.

Lee et al. reported, for the first time, the profiles of the plasma PYGB concentrations in normal pregnancy and in pregnancy complicated with preeclampsia [94]. The plasma PYGB concentrations physiologically increased during normal pregnancy compared to healthy nonpregnancy. The PYGB levels were significantly increased in preterm preeclampsia, but there were no significant differences in patients with term preeclampsia.

A team from Ireland came to this conclusion in their research [95,96,97], showing that the PYGB concentrations had significant variations in normal pregnancy, preeclampsia, and SGA pregnancy. The plasma PYGB of normal pregnancy decreased with the progress of pregnancy from the first to the third trimester. However, Lee et al. reported that there was no difference in the PYGB concentrations with the progress of pregnancy or in pregnancies affected by SGA [94].

In our opinion, the differences in the findings may be due to the different exact test methods used in the studies and the different demographic characteristics of the women included in the studies.

When PE and SGA appeared, the plasma PYGB increased significantly, higher than that of normal uncomplicated pregnancy, indicating that plasma PYGB is a biomarker of uterine and placental dysfunction. That is to say, PYGB can be used as a biomarker for early pregnancy and in the prediction of premature and full-term preeclampsia and SGA onset.

Studies have suggested that there is currently insufficient evidence to support PYGB as a useful diagnostic or prognostic indicator in preeclampsia or SPE. They have found no evidence of subclinical myocardial ischemia with the PYGB concentrations in preeclampsia or SPE, and PYGB had a limited role as a biomarker in hypertensive disorders complicating pregnancy [98]. Due to the small sample size, the research results have certain limitations.

In conclusion, PYGB is the most sensitive marker within 4 h after the onset of chest pain, which can reflect the degree of myocardial injury and determine the early risk stratification of UA patients. PYGB is superior to other biochemical indicators in the early diagnosis of acute coronary syndrome (ACS). The diagnostic effect of CTnI alone is greater than that of CTnI and PYGB, and the combination of PYGB and CTnI does not improve the detection rate of ACS, so PYGB cannot provide independent prognostic information for patients with acute ischemic chest pain. PYGB can also improve the early detection rate of AMI, but the increase in the plasma concentration of PYGB is related to the exercise load/duration, so PYGB cannot be recommended as an independent indicator for diagnosing AMI, and the combination of PYGB and CTnI in the diagnosis of AMI needs further research. PYGB has also been evaluated as a prognostic marker in patients with AMI, which requires further investigation. It is particularly useful for differentiating between AMI and UA within 3 h of admission in patients with chest pain < 3 h. In addition, PYGB is more suitable than CK-MB for the perioperative laboratory monitoring of myocardial ischemia in patients undergoing emergency coronary artery bypass grafting.

Although PYGB has been shown to be a sensitive marker for diagnosing AIS, it can be used as a screening marker for the early diagnosis of suspected stroke. However, high concentrations of PYGB are also present in the human heart, and concurrent myocardial damage may cause PYGB to leak from the myocardium, resulting in an elevated false positive for PYGB.

Plasma PYGB in normal pregnancies decreases with pregnancy progression from the first trimester to the third trimester. The PYGB levels were significantly elevated in patients with the early onset of eclampsia, but not in patients with preeclampsia at term. Due to the small number of cases used in the studies, the many differences in the methods used in the studies, and the fact that the populations studied also have their own distinct geographical characteristics, these results are not universal, so there is currently insufficient evidence to support the use of PYGB as a definitive diagnostic indicator for preeclampsia or SPE, and the role of PYGB as a biomarker in hypertensive disorders complicated by pregnancy is limited.

In summary, we need more research to prove whether PYGB can be used to identify cardiovascular disease. Due to the limited predictive power of PYGB alone for cardiovascular disease, further studies may consider its use in combination with other markers.

## 4. PYGB in Metabolic Diseases

### 4.1. PYGB in McArdle Disease

McArdle disease, also known as glycogen storage disease type V, is an autosomal recessive disorder that is caused by the glycogenolysis disorder of the skeletal muscle due to PYGM defect. Theoretically, any pharmacological treatment that upregulates the PYGB expression in the skeletal muscle could be a potential therapeutic approach for patients with McArdle disease.

In the research of Luna et al., demethylation treatment was performed on the CpG island, the promoter of PYGB, to activate and transcribe the PYGB gene and to promote glycogen catabolism, thereby making up for the deficiency in PYGM. They observed that, with the treatment of sodium valproate (VPA), the expression of PYGB in mouse primary skeletal muscle cells increased and the polysaccharide accumulation was reduced, partially compensating for the accumulation of glycogen caused by the loss of the PYGM expression. Therefore, it was concluded that VPA enhanced the expression of PYGB in vitro and is expected to become a candidate for the treatment of McArdle disease [99].

### 4.2. PYGB in Type 2 Diabetes (T2D)

A study showed that inhibiting mTOR with rapamycin significantly improved the metabolic status and cardiac function in mice with type 2 diabetes (T2D). The rapamycin treatment significantly induced changes in the PYGB proteins [100].

Five insulin resistance (IR)-related SNPs or genes were identified in a pediatric population via a pooled-DNA GWAS. A series of analyses revealed that the most significant one is the SNP rs2258617 within PYGB. The SNP rs2258617 resides within the PYGB gene. PYGB is a new candidate gene involved in the genetic regulation of IR, which provides a target for further basic research on the mechanism of IR and the development of potential new therapies for IR and T2D [101].

### 4.3. PYGB in Late-Onset Pompe Disease (LOPD)

Pompe disease (PD) is a monogenic autosomal recessive disorder caused by biallelic pathogenic variants of the GAA gene encoding lysosomal α-glucosidase; its deletion causes glycogen storage in lysosomes, mainly in the muscular tissue. Genes encoding proteins related to glycogen synthesis and catabolism may be excellent candidates for phenotypic modifiers of PD.

In terms of glycogen catabolism, glucose molecules can be obtained from lysosomes and glycogen in the cytoplasm. In lysosomes, the only enzyme involved is α-glucosidase encoded by GAA, and two kinds of cytoplasmic enzymes are needed to obtain glucose-1-phosphate or free glucose, namely, glycogen phosphorylase and the glycogen-debranching enzyme.

In this study, the potential variation in the genes related to glycogen synthesis and catabolism was found in a group of 30 patients with late-onset Pompe disease (LOPD) via whole-exome sequencing. Among the genes related to glycogen catabolism, PYGB, PYGL, and PYGM, only PYGM is expressed in muscle, which will lead to McArdle disease when it carries biallelic mutations. All of them share very similar z-scores for the NS variants, and the NS variants of PYGB and PYGL exist in a consistent number of cases [102].

At present, there are not enough qualitative/quantitative details to describe the correlation between muscle impairment and genes; even if the function is slightly reduced due to genetic variation, it will affect the phenotype.

## 5. PYGB in Nervous System Diseases

### 5.1. PYGB in Alzheimer’s Disease (AD)

In a study to identify Alzheimer’s disease autoantibodies and their target biomarkers via phage microarray, researchers found that four receptors, including PYGB, could significantly distinguish AD from controls. In AD patients versus control individuals, PYGB was deregulated at the mRNA and protein levels and showed significant overexpression.

In the process of AD, proteins such as PYGB can actually leak out of the brain to promote the humoral immune response, resulting in a higher production of autoantibodies than in healthy individuals. These results indicate that the humoral immune response of AD patients to the PYGB target peptide increases the number of AD blood biomarkers, which can be used for disease detection [103].

### 5.2. PYGB in Amyotrophic Lateral Sclerosis (ALS)

Amyotrophic lateral sclerosis (ALS) is an age-dependent neurodegenerative disease characterized by metabolic dysfunction. It has been found that the regional expression of PYGB in the spinal cords of ALS mice was significantly reduced. As ALS progresses, the regional accumulation of glycogen in the lumbar spinal cords of mice is a result of deteriorating glycogenolysis due to the decreased PYGB. Bioinformatics analysis showed that miRNA-mediated post-transcriptional regulation controlled the expression of PYGB.

MiR-338-3p is a significantly elevated miRNA in the spinal cords of ALS mice that directly targets PYGB. miR-338-3p binds to the 3′UTR of PYGB, thereby inhibiting the expression of PYGB and reducing the glycogen decomposition and subsequent glycogen accumulation. In addition, the inhibition of miR-338-3p by an inhibitor effectively increased the expression of PYGB. Therefore, miR-338-3p provides a potential target for future intervention in the treatment of ALS [104].

### 5.3. PYGB in Depression

Astrocyte glycogen plays a vital role in brain energy metabolism. Glycogen accumulation is closely related to the occurrence and development of depression, and glycogen metabolism disorder participates in the pathogenesis of depression. Depression is a common mental disorder with a high incidence, a high recurrence rate, and high mortality.

The study investigated whether astrocyte PYGB was involved in glycogen accumulation in depression by combining behavioral and genetic methods. Study data showed that the glycogen levels were significantly increased in PYGB knockdown mice, and PYGB knockdown was found to increase the susceptibility to depressive-like behavior in the medial prefrontal cortex (mPFC). In contrast, PYGB overexpression reduced the susceptibility to depression. These results suggested that glycogenolysis reprogramming led to glycogen accumulation in astrocytes. The decreased glycogen degradation resulted in the increased brain glycogen levels in depression. Therefore, glycogenolysis is a potential intervention target for stress, and PYGB plays a key role in glycogen storage. The inactivation of PYGB leads to the dysfunction of glycogen decomposition and, thus, glycogen storage. PYGB contributes to stress-induced depression-like behaviors and is a promising therapeutic target for the treatment of depression [105].

### 5.4. PYGB in Spinal Muscular Atrophy (SMA)

Spinal muscular atrophy (SMA) is an inherited neuromuscular disease, and the majority of cases of SMA are caused by insufficient SMN protein levels due to the loss of function of the survival of the motor neuron 1 (SMN1) gene.

The proteomic profiling of skin fibroblasts from patients and controls was conducted using SWATH mass spectrometry analysis, and it was confirmed via quantitative Western blot that PYGB was significantly increased only in SMA I (severe). The PYGB levels in SMA I fibroblasts were significantly increased by 1.29-fold when compared to controls. It has the potential to serve as a biosignature of SMA I and a biomarker for patient stratification, or for monitoring the efficacy of treatment [106].

## 6. PYGB in Other Diseases

### 6.1. PYGB in Liver Regenerative Medicine

It was concluded that PYGB is a new marker of liver progenitor cells, and that its expression promoted the survival under the low-glucose condition and the cell differentiation of liver progenitor cells [30].

### 6.2. PYGB In Vitro Fertilization (IVF)

The protein composition of the endometrial fluid aspirate (EFA) on the day of embryo transfer with and without pregnancy in in vitro fertilization (IVF) cycles was analyzed via protein omics and confirmed via Western blotting. PYGB was found to be downregulated in non-implantative EFA. PYGB can serve as a candidate to distinguish between the implantative and non-implantative cycles, acting as a potential biomarker of endometrial receptivity or successful implantation. This finding provides a new idea for developing embryo transfer strategies, suggesting the possibility of improving the pregnancy rate by enhancing the embryo culture medium or regulating the composition of endometrial fluid [27].

However, the available data are limited, and we cannot recommend its practical use in these areas until its clinical efficacy is better understood.

## 7. Conclusions and Perspectives

Initially, the function of PYGB was mainly in the catalytic process of glycogen degradation to provide energy for organisms in abnormal or emergency situations. Gradually, it has been discovered that PYGB may play an important role in various diseases, such as cancers, cardiovascular diseases, metabolic diseases, nervous system diseases, etc., and it involves several signaling pathways (Figure 4). Among them, PYGB has been observed to be overexpressed in many types of cancer and tumor cell lines. It is believed to play a role in cancer cell proliferation, differentiation, senescence, and apoptosis. It is expected to be a novel target for the diagnosis and treatment of various cancers. However, changes from basic to clinical research need further functional and underlying-mechanism studies in more detail. The different isoforms of glycogen phosphorylase are upregulated in different types of cancers, but the enzymes are required for glycogen metabolism in all cells in the body. Thus, the inhibition of all glycogen phosphorylase isoforms could result in overwhelming toxicity.

The clinical application of PYGB in cardiovascular diseases has been broadly confirmed in previous studies, and PYGB is an early and sensitive biomarker of many kinds of cardiovascular diseases. However, because its accuracy is not yet clear, it is necessary to adopt strict laboratory research designs and robust standards to conduct prospective studies to determine the clinical usefulness of these tests. In addition, studies on PYGB have been gradually expanded in various diseases, including metabolic diseases, nervous system diseases, and other diseases, over the past decade. However, research progress in these areas has been slow and shallow. With the fast development of high-throughput sequencing and gene-editing technology, we believe there will be more and more researchers devoted to these aspects of research in the future.

Thus, it can be seen that PYGB may have wide application prospects, but there are still many difficulties. Keeping the limitations of these studies in mind, we hope that more studies on its regulatory mechanisms will be conducted in the future. At the same time, we hope this review will help promote the application of PYGB, and we will continue to focus on the research of PYGB in order to obtain more gratifying results.

## Figures and Tables

**Figure 1 cells-13-00289-f001:**
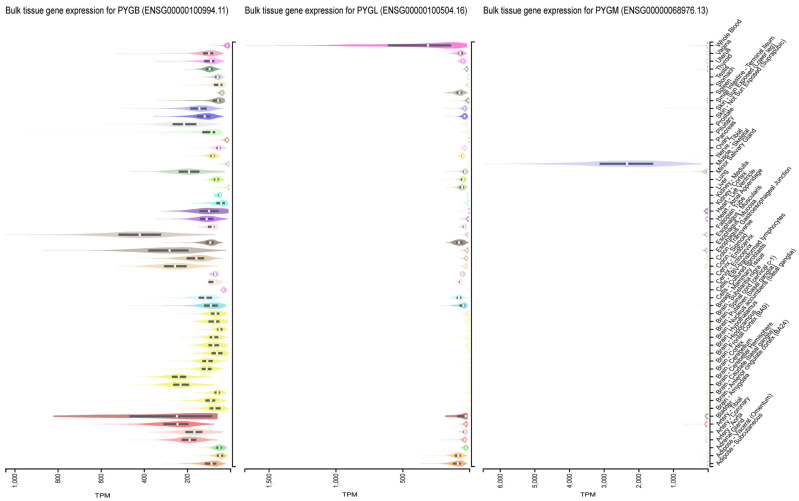
Expression differences in three isoforms of GP in different tissues.

**Figure 2 cells-13-00289-f002:**
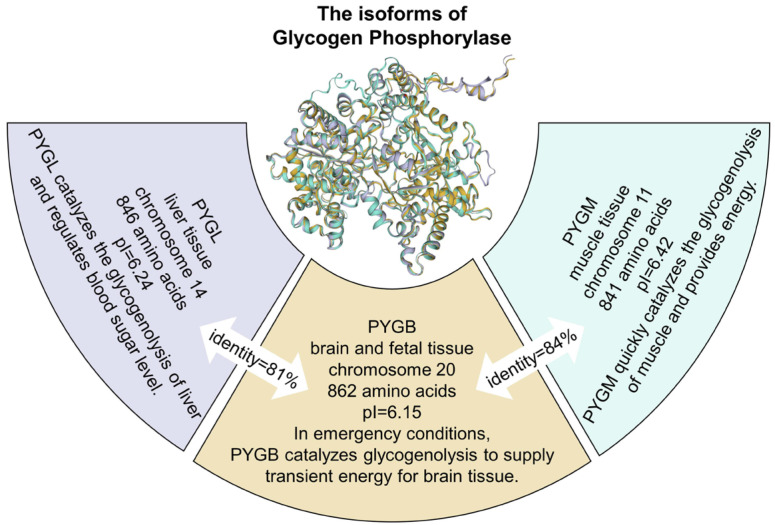
Biochemical and physiological characteristics of glycogen phosphorylase.

**Figure 3 cells-13-00289-f003:**
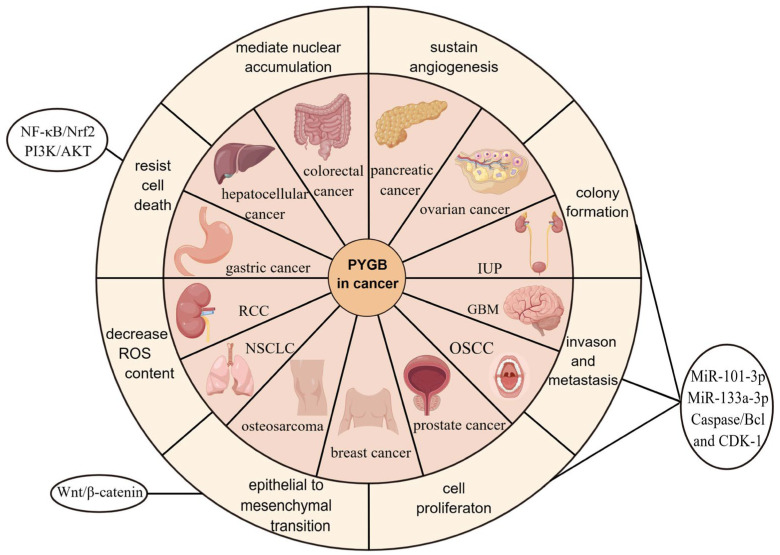
The role of PYGB in cancer and related signaling pathways. By Figdraw. ID:AOYUO3213a.

**Figure 4 cells-13-00289-f004:**
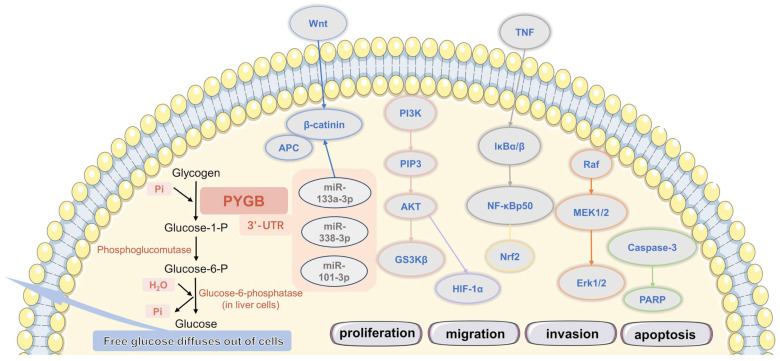
Metabolic activities involved in PYGB, and its related signaling pathways and biological factors.

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
