# Peer review of "Brain-Type Glycogen Phosphorylase (PYGB) in the Pathologies of Diseases: A Systematic Review"

_cells, 2024, doi:10.3390/cells13030289_

Round 1
Reviewer 1 Report
Comments and Suggestions for Authors
· p. 3, line 85: "A systematic review on PYGB is not available up to now." This statement isn't entirely true. The last review I am aware of and see was in 2019 from Mathieu et al., Adv Neurobiology titled "The structure and the regulation of glycol phosphorylases in brain." The authors should reference this review and modify this sentence.
· If they are not aware of it already, the authors should become familiar with this book: Dienel GA, Carlson GM. Major Advances in Brain Glycogen Research: Understanding of the Roles of Glycogen Have Evolved from Emergency Fuel Reserve to Dynamic, Regulated Participant in Diverse Brain Functions. Adv Neurobiol. 2019;23:1-16. doi: 10.1007/978-3-030-27480-1_1. PMID: 31667804.
· Regarding NSCLC, the authors should provide a few sentences to describe the phenomenon of nuclear glycogen metabolism that was originally discovered in the 1940s and recently published in a 2019 Cell Metabolism paper.
o H.D. Chipps, G.L. Duff Glycogen infiltration of the liver cell nuclei Am. J. Pathol., 18 (1942), pp. 645-659
o Sun RC, Dukhande VV, Zhou Z, Young LEA, Emanuelle S, Brainson CF, Gentry MS. Nuclear Glycogenolysis Modulates Histone Acetylation in Human Non-Small Cell Lung Cancers. Cell Metab. 2019 Nov 5;30(5):903-916.e7.
· There are a couple additional papers to reference regarding NSCLC.
o Lei K, Tan B, Liang R, Lyu Y, Wang K, Wang W, Wang K, Hu X, Wu D, Lin H, Wang M. Development and clinical validation of a necroptosis-related gene signature for prediction of prognosis and tumor immunity in lung adenocarcinoma. Am J Cancer Res. 2022 Nov 15;12(11):5160-5182. PMID: 36504901; PMCID: PMC9729905.
o Huang Q, Yang S, Yan H, Chen H, Wang Y, Wang Y. Development and validation of a combined glycolysis and immune prognostic signature for lung squamous cell carcinoma. Front Genet. 2022 Sep 30;13:907058. doi: 10.3389/fgene.2022.907058. PMID: 36246596; PMCID: PMC9561419.
· P. 12, line 511: regarding brain ischemia; the authors should include a summary of and reference the below paper
o Cai Y, Guo H, Fan Z, Zhang X, Wu D, Tang W, Gu T, Wang S, Yin A, Tao L, Ji X, Dong H, Li Y, Xiong L. Glycogenolysis Is Crucial for Astrocytic Glycogen Accumulation and Brain Damage after Reperfusion in Ischemic Stroke. iScience. 2020 May 6;23(5):101136. doi: 10.1016/j.isci.2020.101136. PMID: 32446205; PMCID: PMC7240195.
Comments on the Quality of English LanguageThe English is very good.
Reviewer 2 Report
Comments and Suggestions for Authors
It's a very interesting and vastly underexplored topic in cellular biology. Ever since the discovery of PYG and its isoforms, the dogma has been that these hydrolases catalyze the process of glycogen degradation. The authors' idea to shed light on the isoform B of PYG is excellent, and they have indeed attempted to do so, considering that this field of science is not very up-to-date. One just needs to look at the manuscript's bibliographic references and those published on online platforms.
However, the manuscript still needs to demonstrate the specificity of signal transduction by phosphorylases in general, and by isoform B in particular. Firstly, what is the tissue-specific expression ? All three isoforms exhibit a ubiquitous expression pattern; they are expressed in the immune system, brain, thyroid, for instance. This invalidates or diminishes the tissue-based specificity expression dogma. The table 1 doesn't make sense in this case.
Secondly, what functional differences does PYGBB present compared to M and L. PYGM has been described as an effector molecule of the GTPase Rac1 (Arrizabalaga et al, J biol Chem 2012 Apr 6;287(15):11878-90). This GTPase, from the Rho family, is by definition an oncogene. Wouldn't this tandem be validated to participate in tumorigenic processes, more than PYGBB?
In the cardiovascular chapter and subsequent ones, PYGBB is portrayed as a disease marker, much like many others, and it hasn't had much success. Despite being described in references from its inception to the present day, it doesn't seem to have made much headway. And furthermore, in each subchapter, there's reference to a published article, but it's merely a description of them without any discussion. Regarding the chapter on McArdle's disease, this serves as a prime example of the lack of discussion and correlation between the isoforms. In this specific case, the issue lies in an absolute deficit of functionality of PYGM, initially thought to be exclusive to skeletal muscle. However, the reality suggests that it's not exclusive to skeletal muscle; there are other affected tissues.
This indicates that the absence of PYGM expression affects the expression and activity of other molecules, among which PYGBB is included. Even though it works to degrade glycogen, it doesn't compensate for the lack of PYGM.
Due to all these reasons, despite being a highly appealing topic for Cells' readers, it's not currently ready for publication. It requires a thorough revision and a correlation between the functions of the isoforms.
Round 2
Reviewer 2 Report
Comments and Suggestions for Authors
The modifications made by the authors are acceptable, but before accepting the revision, they should justify the changes that have occurred in the authorship of the manuscript